# The Impact of Climate Change on Aeromedical Retrieval Services in Remote Northern Australia: Planning for a Hotter Future

**DOI:** 10.3390/ijerph21010114

**Published:** 2024-01-20

**Authors:** Simon Quilty, Aparna Lal, Bridget Honan, Dan Chateau, Elen O’Donnell, Jodie Mills

**Affiliations:** 1National Centre of Epidemiology and Population Health, Australian National University, Canberra 2600, ACT, Australia; aparna.lal@anu.edu.au (A.L.); dan.chateau@anu.edu.au (D.C.); 2Medical Retrieval and Consultation Centre, Alice Springs Hospital, Alice Springs 0870, NT, Australia; dr.bridget.honan@gmail.com (B.H.); elenodonnell@gmail.com (E.O.); 3Careflight Northern Territory, Eaton 0820, NT, Australia; jodie.mills@careflight.org

**Keywords:** climate change, environmental heat, aeromedical retrieval

## Abstract

It is known that environmental heat is associated with increased morbidity manifesting as increasing demand on acute care health services including pre-hospital transport and emergency departments. These services play a vital role in emergency care, and in rural and remote locations, where resource capacity is limited, aeromedical and other retrieval services are a vital part of healthcare delivery. There is no research examining how heat impacts remote retrieval service delivery. The Northern Territory (NT) of Australia is characterised by very remote communities with limited acute healthcare capacities and is a region subject to regular extreme tropical heat. In this study, we examine the relationship between aeromedical retrievals and hot weather for all NT retrievals between February 2018 and December 2019. A regression analysis was performed on the number of retrievals by clinical reason for retrieval matched to the temperature on the day of retrieval. There was a statistically significant exposure response relationship with increasing retrievals of obstetric emergencies in hotter weather in the humid climate zone and surgical retrievals in the arid zone. Retrieval services appeared to be at capacity at all times of the year. Given that there are no obstetric services in remote communities and that obstetric emergencies are a higher triage category than other emergencies (i.e., more urgent), such an increase will impede overall retrieval service delivery in hot weather. Increasing surgical retrievals in the arid zone may reflect an increase in soft tissue infections occurring in overcrowded houses in the hotter months of the year. Given that retrieval services are at capacity throughout the year, any increase in demand caused by increasing environmental heat will have broad implications for service delivery as the climate warms. Planning for a hotter future must include building resilient communities by optimising local healthcare capacity and addressing housing and other socioeconomic inequities that amplify heat-related illness.

## 1. Introduction

The Northern Territory (NT) of Australia is a sparsely populated tropical region covering 1.39 million square kilometres, more than twice the size of Spain, characterised by vast distances and very remote communities. These remote communities are largely Indigenous and, from a health perspective, are characterised by extreme poverty, poor-quality and overcrowded housing, very high burdens of chronic disease, and low life expectancy [1,2,3]. In remote areas such as the Northern Territory, there is often limited capacity of local clinics to provide complex emergency care, often in communities where there are high population rates of emergency presentations. Thus, these communities rely on aeromedical retrieval services in emergency health circumstances.

The NT is also a region characterised by a very hot climate, with the northern ‘Top End’ experiencing tropical and humid weather and the southern NT, Central Australia, defined by a hot arid climate. The impact of climate change is already being experienced with dramatic increases in the duration and extent of heat seasons and heat events, and deeply concerning longer-term forecasts of still further substantial increases in extreme heat [4]. There are two small cities in the NT—Darwin at 12° south in the tropics and Alice Springs in Central Australia at the Tropic of Capricorn (23° south).

Excessive environmental heat, including heatwaves and seasonal summer weather, has been shown to cause significant morbidity and mortality [5,6]. This leads to increased burden on health services and higher healthcare costs [7], including a greater demand on emergency retrieval services and emergency department presentations [8]. For instance, it is known that heatwaves lead to an increase in hospital admissions [9,10] and that this in turn leads to an increase in the utilisation of ambulance and retrieval services and delays in retrieval [8,11].

The effects of heat on morbidity and mortality are influenced by a complex interplay between environmental, socioeconomic, sociocultural, and individual physiological vulnerabilities [12]. Several comorbidities, such as older age and specific medical conditions, are known to increase the risk of heat-related illnesses [13]. For example, environmental heat is associated with increased hospitalisations for cardiovascular, respiratory, metabolic, and renal diseases [14]. Rates of trauma, injury, and psychiatric illnesses also tend to rise during hotter weather [15,16]. Moreover, hot weather negatively affects maternal, foetal, and neonatal outcomes [17]. In regions like the Northern Territory, characterised by vulnerable populations, limited local healthcare capacity, and vast distances to servicing hospitals, it is imperative that health services take into account the impacts of climate change and how such adverse health outcomes of this complex interplay will be addressed.

In rural and remote areas, the costs and carbon footprint of aeromedical retrieval are amplified due to the need for air transport, yet there is limited research and understanding of how heat impacts these services in such regions [18]. Given that climate change is already substantially increasing environmental heat in already very hot climate zones, understanding the relationship between heat vulnerability and healthcare utilisation during heat events is crucial for effective planning and preparation for a hotter climate. It enables the development of public health programs to protect the most vulnerable individuals, particularly in rural and remote areas characterised by logistical challenges and limited access to emergency healthcare services [19].

The objective of this study is to analyse the association between temperature and aeromedical retrievals in two remote regions defined by different aeromedical service providers and located in different climate zones (tropical and arid) of northern Australia. Additionally, we aim to investigate the specific reasons for aeromedical retrieval in order to gain a more nuanced understanding of vulnerabilities in the region that may provide a better understanding of the types of acute care service demands associated with hot weather.

## 2. Materials and Methods

This study received ethics approval from the Northern Territory (NT) Department of Health Human Research Ethics Committee (HREC 2020-3822).

### 2.1. Setting

There are two free-to-patient government-funded aeromedical retrieval services in the NT—Careflight in the humid Top End region and Royal Flying Doctor Service (RFDS) in the arid Central region. Careflight services operate predominantly from Darwin and feed into Royal Darwin Hospital (RDH), although there is a smaller retrieval base in Nhulumbuy that also feeds locally into Gove District Hospital and RDH. RFDS feeds only into Alice Springs Hospital (ASH). The climate of the two regions serviced by these retrieval services is hot and humid in the Top End (Darwin) and hot arid desert climate in the Central Australian (Alice Springs) region. The data cover all aeromedical retrievals in the Northern Territory from February 2018 to December 2019.

### 2.2. Participants

An episode of retrieval was defined as an aeromedical aircraft responding to a referral for emergency retrieval and collecting the remotely located patient. Episodes were excluded if there were incomplete data regarding the pick-up location or if no reason was recorded for retrieval.

### 2.3. Variables

The retrieved data include the date, retrieval locations (from and to), patient’s age and sex, reason for retrieval (recorded in medical shorthand by the medical retrieval coordinator), triage category (ranging from 1 to 5 in the humid zone by Careflight and 1 to 7 in the arid zone by RFDS, with Careflight and RFDS using different triaging definitions; however, for both services, triage category 1 is the highest urgency requiring immediate action; categories 2, 3, and 4 represent decreasing urgency, being approximately equivalent between services, where category 4 is considered semi-urgent; and triage category 5 and over is non-urgent), and whether a doctor accompanied the retrieval team. The reason for retrieval was coded from retrieval service ‘reason for referral’ data electronically entered at the time that the referral was made by the triaging doctor. We used a machine learning algorithm on this ‘reason for referral’ data [20] to code all retrievals into medical, surgical, trauma, psychiatric, and obstetric emergencies on the basis of the specialist care type likely to be received once arriving at the hospital. Coding based on the ICD (International Classification of Disease) was not possible due to a lack of diagnostic information prior to hospital arrival as referrals are made in remote locations with minimal diagnostic capacity and sometimes by non-clinician referrers; thus, records of ‘reason for referral’ could not be more granular from a coding perspective.

### 2.4. Data Sources

The aeromedical retrieval data were obtained from Careflight Australia in the Top End and from the Royal Flying Doctor Service (RFDS) in Central Australia. These two services cover the entire Northern Territory using aeromedical retrieval ambulances.

Temperature data recorded by the Australian Bureau of Meteorology (mean temperature) on the day the retrieval was tasked were used to define temperature for each retrieval episode (Figure 1). For RFDS retrievals in Central Australia, the temperature from Alice Springs was used, while for Careflight retrievals in the Top End, the temperature from Darwin was used.

Heat-attributable morbidity and mortality typically peak at a 3-day lag [21]. Thus, temperature was defined as the mean over the preceding three days. The 3-day lag mean temperature on retrieval day was divided into temperature quartiles based on the weather records in Darwin (humid zone) and Alice Springs (arid zone), representing the coolest 25% of days to the hottest 25% during the study period. It was assumed that the temperature quartiles at the location of retrieval were the same as those at their corresponding destinations (Darwin for the humid zone locations and Alice Springs for the arid zone locations).

### 2.5. Data Analysis

The population at each retrieval location was estimated based on the 2016 population census data from the Australian Bureau of Statistics [22]; however, these data did not account for seasonal population variation. Remote areas of the Northern Territory have a small baseline population, and there is a substantial influx to those towns and regions offering tourist destinations in the cooler months from May until September, an influx that is known to impact healthcare and retrieval services in the NT. A monthly population factor was applied to adjust for seasonal population fluctuations in towns along tourist routes, considering the significant tourism and seasonal worker industry during the cooler “dry” season. However, this adjustment was not applied to more remote communities that do not experience such population fluctuations. The population factor was calculated using sales volumes per month for a 12-month period of a universally consumed product sold in the only supermarket in Tennant Creek, ranging from 1.0 in the hottest month (December) to 1.50 in the coolest month (August), and reflecting a 50% increase in population to these areas during August. The population factor was then multiplied by the baseline population recorded in the 2016 census data by month to account for these seasonal population fluctuations.

An analysis was conducted using STATA v.18 software examining mean 3-day lag temperatures by reason for retrieval, performed separately for the northern humid zone (retrievals to Darwin) and the cooler arid zone (Central Australia) due to the separate aeromedical service providers and their distinct climates [11]. Poisson or negative binomial regression analyses were conducted on the count of retrievals for each reason for referral, with the corresponding population in the region included as an offset. Where the variance was greater than the mean, overdispersion was assumed and negative binomial regression was used. The mean temperature for the region was entered as an exposure variable, with a 3-day lag.

To determine whether there is an increase in higher-urgency triages occurring due to hot weather, an analysis of the triage category vs. temperature quartile was performed in each region by reason of retrieval using a chi square test of association to facilitate interpretation of the results.

## 3. Results

### 3.1. Climate

Daily temperatures in both Darwin and Alice Springs were hot, with a smaller range of variation in the tropical city of Darwin and very hot summers and brief periods of cooler weather in Alice Springs (Figure 1).

### 3.2. Retrievals

In the humid zone, there were a total of 6200 retrievals, while the arid zone had 3214 retrievals, with retrieval rates consistent with the baseline populations of each region. Medical reasons accounted for the majority of retrievals (3079 for humid and 1654 for arid), followed by surgical (1243 for humid and 685 for arid), trauma (973 for humid and 532 for arid), obstetric (419 for humid and 222 for arid), and psychiatric (486 for humid and 122 for arid) retrievals across both regions (Table 1). Notably, the number of psychiatric retrievals per population was approximately half in the arid zone compared to the humid zone (Figure 2). There was no difference in overall retrieval rates in each temperature quartile in either the humid or arid zones. Medical retrievals per population were slightly higher in each temperature quartile in the arid zone compared to the humid zone. Psychiatric retrievals were nearly twice as frequent in the humid zone compared to the arid zone across all four temperature quartiles. Trauma, obstetric, and surgical retrievals exhibited similar population rates for all temperature quartiles.

### 3.3. Triage Category by Temperature

The triage category is selected based on the clinical urgency of the retrieval referral. The mean triage referral per temperature quartile provides insight into whether hot weather was associated with a greater number of higher urgency retrievals as a surrogate of heat-induced systems stress. There was no statistically significant difference between triage category of retrieval by temperature (Figure 3). In the humid region, obstetric emergencies had lower numbers for their triage categories (i.e., more urgent emergency retrievals) than other reasons for retrieval, and surgery had higher numbers for their triage categories (i.e., less urgent). The same was true for the arid region, which also had lower numbers for the triage categories (i.e., more urgent) for trauma retrievals.

### 3.4. Regression Analysis of Retrievals by Temperature Quartile

There was a statistically significant dose–response increase in obstetric retrievals from the lowest quartile to the highest quartile in the humid zone (IRR 1.49 for hottest quartile compared to coldest, *p* = 0.005) (Table 2). There was no association with temperature for obstetric retrievals in the arid zone (Table 2).

There was a statistically significant increase in surgical transfers during the hottest temperature quartile in the arid zone (IRR 1.64 for hottest quartile compared to the coldest, *p* = 0.032 Table 2). There was no association with temperature for other reasons for retrieval in the arid zone.

## 4. Discussion

Our analysis revealed several key results regarding the association between temperature and aeromedical retrievals in the remote region of northern Australia. Firstly, a dose–response relationship was found between hotter weather and obstetric retrievals in the humid zone. Pregnant women and foetuses are particularly vulnerable to environmental heat, with evidence linking heat exposure to increased rates of early and preterm birth, stillbirth, and low birth weight [23]. In the NT, pregnant women experience prolonged exposure to very hot temperatures, and our study revealed a statistically significant dose–response relationship between hotter weather and obstetric emergencies and retrievals.

Additionally, we observed a statistically significant increase in surgical retrievals during hotter weather in the arid zone. Although not statistically significant, there were more surgical retrievals in warmer weather in the humid zone. Remote communities in the NT experience high rates of soft tissue infections, which can be exacerbated by hot weather conditions [24]. Overcrowded housing due to extreme environmental heat forces people to congregate, increasing the risk of infectious diseases such as scabies, streptococcus, and staphylococcus [2,25]. The combination of overcrowding and hot weather may contribute to the increase in surgical-related aeromedical retrievals.

There was a non-significant decrease in trauma-related retrievals during hotter weather in our study period. The NT experiences a significant influx of tourists during the cooler months, leading to increased traffic on high-speed highways. Transport-related deaths are 2.5 times more common during these cooler periods in the NT due to this increased high-speed traffic [26] compared to the hotter periods. However, hotter weather is also associated with a rise in violence and suicide-related events, which may partially offset any other decrease in trauma-related retrievals associated with hotter weather and reduced traffic.

We did not find significant associations for medical, trauma, or psychiatric retrievals across the temperature range, which is discordant with known evidence demonstrating worse health outcomes associated with heat events [12]. There are a limited number of aeromedical retrieval assets available in the Northern Territory (NT) and services operate at full capacity for much of the year. This is supported by the fact that in both humid and arid regions, the total number of retrievals across the four temperature quartiles do not differ. It is possible that tasked retrievals may not fully reflect the true demand for retrieval services. Thus, if there are particular population heat vulnerabilities such as obstetrics and surgical ones that result in higher retrievals in hotter weather, this will come within the confines of service capacity, meaning that retrieval services will not be able to meet increased demand. A possible explanation for the consistency of the rates in a system operating at capacity is that the severity of cases might change, rather than the number or rate. However, triage category, i.e., level of acuity of emergency, was not related to temperature quartile for any reason or in either climate zones. It was noted that obstetric retrievals in both regions had lower (i.e., more urgent) categorisation, and surgical reasons had higher (less urgent) category numbers, and in the humid zone, trauma also had lower (more urgent) category numbers. A reason for the more urgent obstetric retrievals is the absence of any remote capacity to be able to provide clinical care to women in childbirth. A reason that surgical retrievals have lower urgency is most likely that soft tissue infections make up a large component of these retrievals, are clinically diagnosable, and are usually non-emergent but require surgical and anaesthetic services that cannot be provided remotely. There is a discrepancy between trauma (lower urgency in the humid zone and higher urgency in the arid zone). This may reflect a discrepancy in injury type (for instance, a high speed MVA will require a different triage than a suspected fractured arm). Given that aeromedical retrieval services are almost constantly at full capacity, if one clinical reason for retrieval increases in hotter weather, then the entire service capacity can be compromised. For instance, given that obstetric retrievals have a lower (more acute) triage category number, the increase in retrievals for this category will result in decreased capacity for other retrieval categories.

There was a noticeable discrepancy in total number of psychiatric retrievals in the humid zone that were twice as high as in the arid zone. The reasons for such a discrepancy are unlikely to be associated with climate zone. It is more likely that there is a discrepancy between the diagnosis and acute management of mental illness between remote and acute care health services in the Top End compared to the Central region. This may possibly also reflect cultural differences in manifestation of acute mental illness between distinctly different and unique Top End vs. Central First Nations societies. This phenomenon deserves further attention by health service providers within the NT.

Health vulnerabilities differ across climate zones, reflecting the interplay between climate and sociocultural, infrastructural, and adaptive individual and population characteristics [12,27]. Despite both climate zones described in this study being very hot, the temperature ranges across the northern humid zone are rarely cold, and there is high humidity. In comparison, the temperature range in the arid zone includes brief cold periods and mostly very low humidity. Humidity, or its lack thereof, primarily affects human health due to its role in heat stress and hydration. The role of humidity and health is complex and climate-zone-dependent [28]. Given the strong correlation between temperature and humidity in the humid zone and persistently low humidity in the arid zone, this study did not examine the role of humidity in the rates of retrieval outcomes. Regardless, climate zones should be taken into account when assessing the impact of climate on health and health services as each climate zone will manifest upon population health in different ways. More research around heat and the interplay with humidity on health is needed.

There are numerous implications for local and broader adaptation plans relating to the findings of this study. Firstly, investments in rural and remote community resilience and clinical capacity will offset the need for retrieval services. Aeromedical retrieval services are very costly and have large carbon footprints, and all effort should be made to reduce the reliance of rural and remote-living people on such services wherever possible. Targeting remote clinical capacity, for instance through the use of telehealth, by providing local options for safe childbirth and by investing in greater basic surgical skills in Indigenous Health Practitioner training may reduce aeromedical retrieval demand. And, whilst strategically investing in clinical capacity in rural and remote settings is one way of achieving less reliance on aeromedical retrieval services, addressing the wide socioeconomic and political inequities between remote living and Indigenous Australians and their urban counterparts could offer even greater long-term improvements in health and wellbeing and may offer an even better utilitarian investment to secure these communities against the challenges posed by climate change.

Several limitations should be considered when interpreting our findings. Firstly, the study relied on retrospective observational data, which may be subject to inherent biases and limitations in data completeness. Additionally, the analysis focused on aeromedical retrievals and may not fully capture the overall demand for healthcare services during hotter weather. Furthermore, our study period covered only two years, which may not capture long-term trends and variations. The categorisation of temperatures into four quartiles may not have been sensitive enough to distinguish periods of extreme heat from baseline temperatures. Lastly, the analysis was limited to specific regions in the Northern Territory, and the generalizability of the findings to other geographic areas should be approached with caution.

## 5. Conclusions

This study reveals a significant association between temperature and aeromedical retrievals in the remote regions of northern Australia. These findings highlight the vulnerability of certain population groups and the need for targeted interventions to mitigate the impact of climate-related conditions on healthcare demand in these regions, and the challenges that will arise on service delivery with increasing duration and intensity of hot weather.

The results of this study demonstrate some important areas for future health service planning. From our study, it is apparent that obstetric services in the humid region of the Northern Territory will have increased demand in hotter weather and that this will impact retrieval services and potentially compromise non-obstetric patient care. Given that there are two remote hospitals in the humid region (Katherine and Nhulumbuy), ensuring that these hospitals have obstetric capacity in the hottest months of the year may mitigate the increased demand on retrieval services.

The impending climate crisis requires urgent research to quantify and describe the resultant impact on demand for health services. This is particularly important for socioeconomic regions that have poorer access to healthcare services and worse healthcare outcomes from an equality perspective. Furthermore, efforts should be made to identify interventions to reduce aeromedical retrievals in order to mitigate against carbon emissions associated with fuel use, such as expanding and enhancing clinical service capacity in remote Australian hospitals significantly reducing the demand for aeromedical retrievals [29].

The elephant in the room of this paper needs to be declared. Aeromedical retrieval services are intensely carbon-emitting. All human activities including healthcare-related life-saving services that leave large carbon footprints need to be intensely scrutinised to determine whether the future costs of such practices would be tolerable for future generations. Medical professions, like all others, have to start taking real, tangible action to reduce carbon emissions associated with our profession. From the lead author’s perspective, after nearly two decades of remote healthcare experience in the NT, there are still far too many aeromedical retrievals tasked in the NT on the basis of individual clinical risk mitigation without greater acknowledgement of the costs and harms of burning thousands of litres of jet fuel when other clinical/logistic approaches would probably have sufficed. We must start to walk the walk and do so with clinical courage and wisdom.

## Figures and Tables

**Figure 1 ijerph-21-00114-f001:**
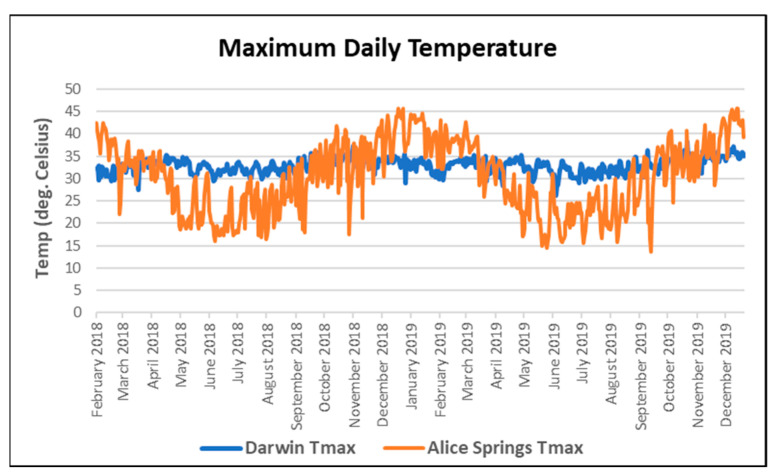
Daily maximum temperatures for Darwin and Alice Springs, demonstrating the extent of hot weather and differences between the two climate zones.

**Figure 2 ijerph-21-00114-f002:**
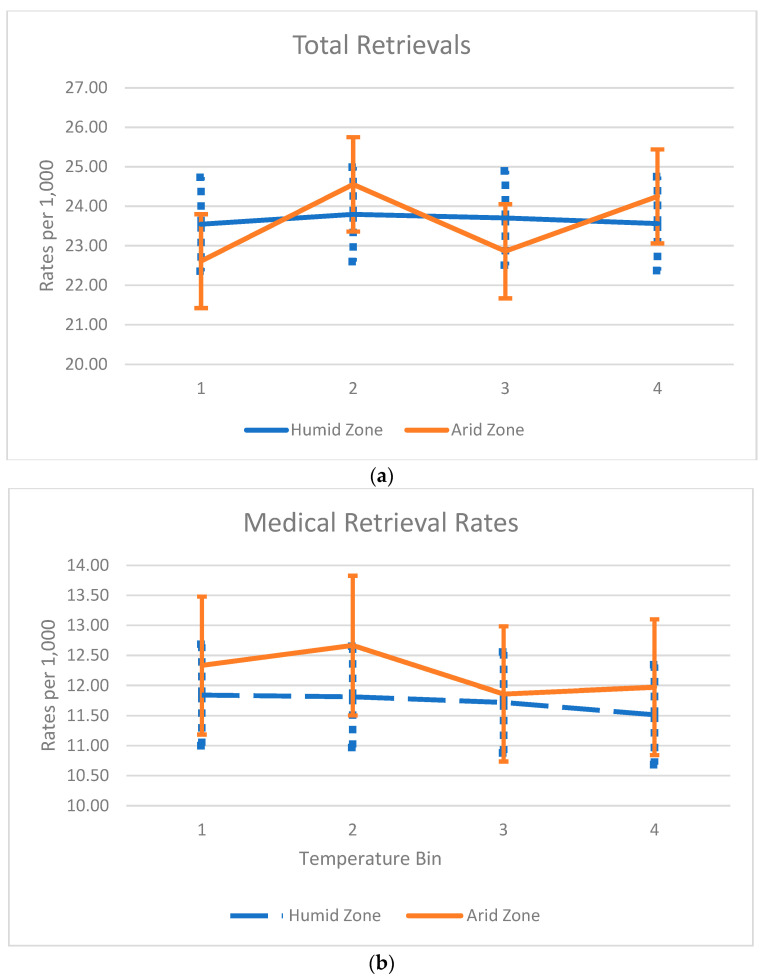
(**a**–**f**) Rates of retrievals per 1000 persons over the study period (quartile 1 being the coldest 25% of days and quartile 4 being the hottest 25% of days in the year).

**Figure 3 ijerph-21-00114-f003:**
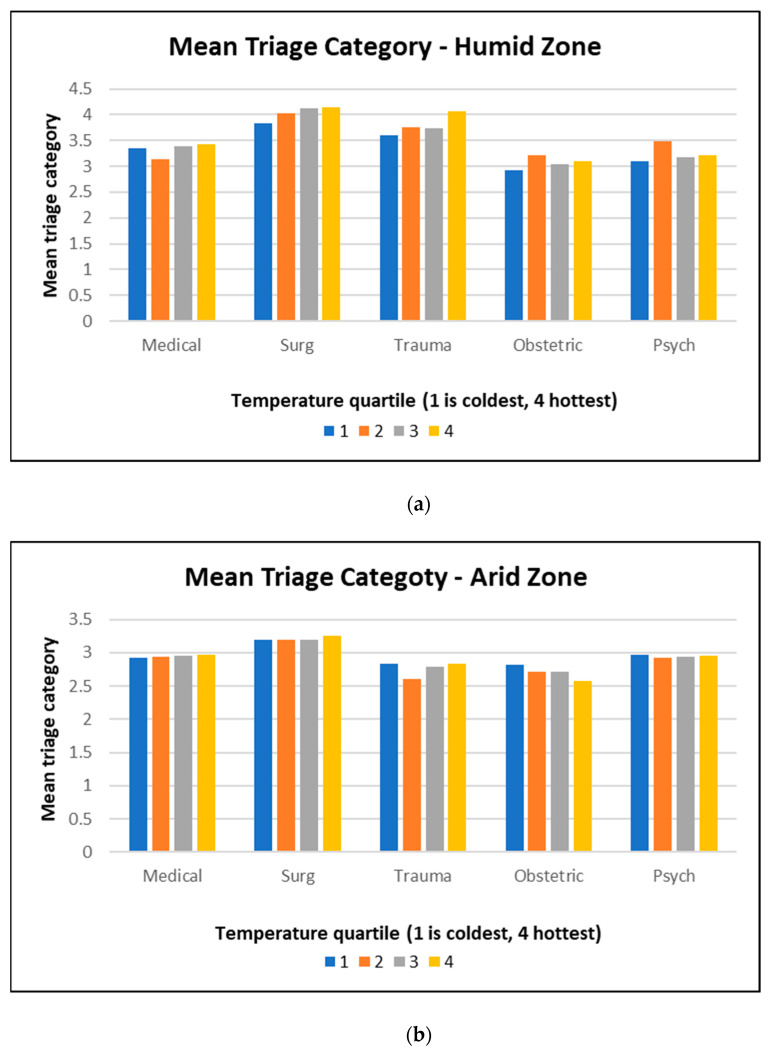
Mean triage category of retrieval by temperature, (**a**) humid and (**b**) arid regions (lower triage category means higher acuity emergency response), noting that Careflight and RFDS have slightly differing categories and thus are not directly comparable between regions.

**Table 1 ijerph-21-00114-t001:** Number of retrievals for each climate zone, by reason for retrieval.

Reason for Retrieval	Humid Zone	Arid Zone
Medical	3079	1654
Surgical	1243	685
trauma	973	532
Psychiatric	486	121
Obstetrics	419	222
Total retrievals	6200	3214

**Table 2 ijerph-21-00114-t002:** Incidence rate ratio (1 = coldest 25% of 3-day lag periods, 4 = hottest) for climate zone and reason for retrieval. ** indicates analyses using negative binomial regression. All other analyses use Poisson regression.

Incidence Rate Ratio by Temperature Bin for Each Climate Zone											
		Humid Zone					Arid Zone				
	Temp. Bin	IRR	CI			*p* Value	IRR	CI			*p* Value
Total	1	1					1				
	2	1.011	0.941	-	1.085	0.771	1.086	0.987	-	1.194	0.089
	3	1.007	0.937	-	1.081	0.856	1.011	0.918	-	1.114	0.824
	4	1.001	0.932	-	1.074	0.985	1.072	0.975	-	1.180	0.151
Medical	1	1					1	**			
	2	0.997	0.901	-	1.1	0.959	1.055	0.671	-	1.657	0.817
	3	0.989	0.894	-	1.09	0.837	1.029	0.652	-	1.623	0.902
	4	0.972	0.879	-	1.08	0.587	0.975	0.619	-	1.536	0.914
Surgical	1	1					1	**			
	2	0.969	0.823	-	1.14	0.708	1.221	0.77	-	1.936	0.396
	3	1.109	0.947	-	1.3	0.198	1.3	0.82	-	2.06	0.265
	4	1.055	0.899	-	1.24	0.514	1.637	1.043	-	2.568	0.032
Trauma	1	1					1				
	2	1.031	0.868	-	1.23	0.725	1.097	0.876	-	1.373	0.422
	3	0.878	0.734	-	1.05	0.157	0.959	0.76	-	1.209	0.722
	4	0.855	0.714	-	1.02	0.089	0.89	0.702	-	1.129	0.334
Psychiatric	1	1					1				
	2	1.034	0.802	-	1.33	0.795	0.931	0.551	-	1.572	0.789
	3	0.957	0.739	-	1.24	0.741	1.241	0.761	-	2.024	0.386
	4	1.017	0.788	-	1.31	0.896	1.138	0.691	-	1.874	0.612
Obstetric	1	1					1				
	2	1.179	0.881	-	1.58	0.268	1.255	0.869	-	1.812	0.226
	3	1.262	0.948	-	1.68	0.111	0.98	0.664	-	1.448	0.921
	4	1.488	1.129	-	1.96	0.005	1.196	0.825	-	1.725	0.345

## Data Availability

The data sets presented in this article are not readily available because they contain patient-level identifiable sensitive health data. Requests to access this data should be addressed to the NT Department of Health Human Research and Ethics committee referencing this study and directed also to Careflight Australia and the Royal Flying Doctors Service.

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
