# Peer review of "The Impact of Climate Change on Aeromedical Retrieval Services in Remote Northern Australia: Planning for a Hotter Future"

_ijerph, 2024, doi:10.3390/ijerph21010114_

Round 1

Reviewer 1 Report

Comments and Suggestions for Authors

Comments on the Quality of English Language

There are minor grammatical errors and typos that should be corrected before publication.

Author Response

The reviewer recommends that our paper should provide a more comprehensive review of health care planning in respect to climate change requires more elaboration. Our intention was to review the association between heat and aeromedical retrieval services in rural/remote settings, which we have previously reviewed and published a scoping study referenced in this paper [O'Donnell E HB, Quilty S, Schultz R. The Effect of Heat Events on Prehospital and Retrieval Service Utilization in Rural and Remote Areas: A Scoping Review. Prehospital and Disaster Medicine. 2021; 36:782-7] . When writing a focused research paper on climate change and health, there is great depth and breadth in both direct and indirect health impacts all of which are population specific. We have chosen to focus entirely on the very under-researched issue of aeromedical retrieval in rural and remote locations. 

  1. Edited as suggested
  2. Clarified as recommended with addition of reference on life expectancy
  3. The reviewer suggests exploration of 'deeper processes by which heat affects certain illnesses or obstetric outcomes'. The pathophysiological processes behind heat and illness are complex and often poorly understood and best addressed by human physiological research, and we have chosen to avoid deeper exploration of these processes as it would expand the focus beyond the objectives of this paper. We suggest that the references we have provided in this paragraph provide the reader with avenues to further explore this complex and important area of research. We agree with the reviewer that "an explanation of why these effects are particularly critical" is worthwhile, and have described in more detail in this paragraph (addition of lines 68-72).
  4. We have referenced of our scoping review which demonstrates the paucity of research regarding aeromedical retrieval and heat  [O'Donnell E HB, Quilty S, Schultz R. The Effect of Heat Events on Prehospital and Retrieval Service Utilization in Rural and Remote Areas: A Scoping Review. Prehospital and Disaster Medicine. 2021; 36:782-7]. Our intention is to focus on the impacts of heat and rates of aeromedical retrieval. Issues impacting rural and remote communities are frequently undocumented due to an unconscious academic bias against rural and remote communities, as attested by the body of international literature on heat and medical retrieval services which is composed of multiple papers exploring the impact of heat on ambulance services in urban locations, and minimal research like ours exploring retrieval services in rural and remote regions.  
  5. We have now included this in lines 45-49 by expanding this paragraph.
  6. We have further clarified the difference between triage categories between services providers in the text (lines 112-116) and in figure 2 description.
  7. The medical shorthand used by retrieval clinicians at the time of the referral provides abbreviated description of the clinical situation. Given that these referrals are made remotely, occasionally by non-clinical referrers, a more robust diagnostic coding cannot be used. We have included a further description of the rationale for this coding from lines 117-126.
  8. The data analysis description has been expanded as recommended by reviewer 1 (lines 143-172).
  9. The Northern Territory experiences substantial seasonal population variation due primarily to tourism in the cooler months of the year. Such regional variation is undocumented but well-known. This seasonal variation is only experienced by towns/regions with high tourism, and most remote Indigenous communities do not experience such seasonal fluctuations. We have further detailed our approach to this quandary in text added to lines 149-159.
  10. We concur with reviewer 1 and have further described the reasons for including this descriptive in interpreting the results from line 169-170 and lines 213-216.
  11. We agree with reviewer 1's concerns particularly in relation to potential cost to patient which would influence uptake and have now highlighted that the service is free to patients and government funded (line 93). The retrieval databases used did not include any demographic information in relation to the individual patient apart from age and sex, and even this age data was considered often to be a 'best guess' by the referring person or clinician. Given that the focus of our study was heat impacts on service delivery we decided not to further explore subgroup analysis. 
  12. We have corrected 'figure' to 'table'. We have included IRR, p values and confidence intervals which we feel is sufficient to describe the analysis without compromising the clarity of description and is consistent with standard poisson/negative binomial model reporting in medical literature.
  13. Reviewer comments noted and now described - lines 165-167.
  14. There is a complex relationship between temperature and humidity that we agree with reviewer 1 deserves further analysis. Given the close relationship between temperature and humidity in tropical regions, and a marker of total atmospheric energy, the relationship between dry bulb temperature and humidity is beyond the scope of this paper, however we now highlight this complex interplay in lines 304-315.
  15. Local and international research has demonstrated that rates of unintentional injuries, motor vehicle accidents and violence increase in hotter weather. We have interpreted statistically non-significant results, i.e. overall rates of trauma that do not change throughout the year, with patterns of seasonal population flux to explain these findings and feel that this paragraph is deserving of being included in discussion.
  16. The population rates of psychiatric retrievals were approximately 1.8/1,000 in the Top End and 0.9/1,000 in the Centre (figure 1). Population rates for all other categories were approximately the same. As described in Data Analysis (line 144) population was estimated through the 2016 National Census data.
  17. We concur with reviewer 1 that Discussion warrants deeper exploration of the issue of climate zone and humidity and have added an extra paragraph (lines 304-315).

Reviewer 2 Report

Comments and Suggestions for Authors

The manuscript exhibits a moderate level of structure. However, I have some suggestions for revisions:

Abstract

-       Enhance the presentation of the statistical analysis power to underscore its significance.

Introduction

-       Line 38-39: Clarify the connection between indigenous people and chronic disease.

-       Include the temperature data for each season.

Materials and Methods

-       Include background information on the study area, including latitude and longitude coordinates.

Setting

-       In line 88, please verify and ensure consistency with the study period mentioned in the abstract.

Variables

-       Please provide an explanation of ICD codes.

Data Analysis

-       Please specify the name of the software used for data analysis.

Results

-       Introduce subtopics or sections within the Results to help readers understand the findings clearly.

-       Ensure that information presented in the Table 1 is not repeated in the text.

-       Figure 1, provide subfigures (a) to (f) and explain the rationale behind different scales for the y-axis in each figure.

-       In Figure 2, elucidate the reasoning behind the utilization of distinct y-axis scales for each subfigure. This explanation is essential for readers to comprehend the specific considerations or characteristics of the data represented in each graph.

-       In line 181, clarify whether the information being referred to is associated with a figure or a table.

-       Page 8, can you explain the meaning or significance of the symbol **”?

Discussion

-       Examine the implications of the national adaptation plan on acute care services.

-       Offer recommendations for both the agency and areas of potential further research.

Conclusions

-       Avoid redundancy with the Results section by presenting a concise and precise summary.

Kindly adhere to the journal template, which includes various sections such as acknowledgments, author contributions, and others.

Author Response

We thank reviewer 2 for their feedback and have made the following changes:

Introduction

  •       Line 38-39: Clarify the connection between indigenous people and chronic disease.

We have increased the description and included the social drivers of these health outcomes, and included two extra references (lines 41-43).

  •       Include the temperature data for each season.

We concur with reviewer 2 that a better description of the prevailing climate would contextualise this research and have included an extra figure (1) demonstrating the variability in temperatures between the humid and arid zone over the study period.

Materials and Methods

  •       Include background information on the study area, including latitude and longitude coordinates.

We have added description of latitude (lines 52-53).

Setting

  •       In line 88, please verify and ensure consistency with the study period mentioned in the abstract.

Corrected in the abstract.

Variables

  •       Please provide an explanation of ICD codes.

ICD (International Classification of Disease) codes were not used as the data was not granular enough. Clarified in line 116.

Data Analysis

  •       Please specify the name of the software used for data analysis.

Corrected. STATA now referred to in text, line 160.

Results

  •       Introduce subtopics or sections within the Results to help readers understand the findings clearly.

Subheadings now included in methods and results.

  •       Ensure that information presented in the Table 1 is not repeated in the text.

In medical research, repeating the text described in tables is often the preference from our experience and this has been done on purpose. We will be guided by IJERPH editors as to the journal's preferred style.

  •       Figure 1, provide subfigures (a) to (f) and explain the rationale behind different scales for the y-axis in each figure.

Subfigures have been lettered as suggested. The different scales for the y axis are rates per 1,000 population as described in the figure text, line 211-212. Scales are best fit for each category depending on baseline rates.

  •       In Figure 2, elucidate the reasoning behind the utilization of distinct y-axis scales for each subfigure. This explanation is essential for readers to comprehend the specific considerations or characteristics of the data represented in each graph.

These figures have been re-done to clarify as per this feedback.

  •       In line 181, clarify whether the information being referred to is associated with a figure or a table.

Clarified by figure being re-done. 

  •       Page 8, can you explain the meaning or significance of the symbol “**”?

The symbol ** is referenced in the figure description in line 240.

Discussion

  •       Examine the implications of the national adaptation plan on acute care services. Offer recommendations for both the agency and areas of potential further research.

Two extra paragraphs added, lines 304-330.

Conclusions

  •       Avoid redundancy with the Results section by presenting a concise and precise summary.

We agree with reviewer 2, sentence deleted in first paragraph of conclusion.

Reviewer 3 Report

Comments and Suggestions for Authors

Introduction:

The introduction lays the groundwork well for the purpose of the article. It flows well the description of the local context, an accounting of the larger problem of heat related illness as a human health hazard, and finally the relationship between aeromedical retrieval and this interplay of problems.

Methods:

Ethics approval is clearly listed in the methods.

The methods are clear and the data sources well delineated and justified.

Results:

Results are well described with adequate graphical representation of results. The breakdown of data by arid and humid zone in different temperature “bins” is easily interpreted.

Discussion:

Authors do a good job of accounting for unexpected results like the decrease in trauma related retrievals during hotter weather and offering possible explanations.

The authors also provide a nuanced accounting for the consistency of the retrieval rates due to a flight system that already operates at capacity regardless of the time of year. While this is briefly touched on in the conclusion, this may be an opportunity for authors to discuss the need for health system adaptation to meet the expected rising demand for healthcare services with rising temperatures.

I was surprised by the lack of significant associations with medical, trauma and psychiatric retrievals. I see that the authors describe that the systems are operating at capacity, but it may be helpful to provide context from other data sets (public health data from the region, hospital admission rates) to show that this is likely actually discordant with actual demand for these patient groups.

The limitations paragraph touched on the fact that aeromedical retrievals, particularly for a system already operating at capacity, may not capture the demand for healthcare services during hotter weather. This would be another opportunity to highlight how this is discrepant from other data sets.

Conclusion:

The conclusion does a good job of highlighting how their data support the need for planning that is sensitive to climate related health hazards like heat. I appreciate that the authors suggest capacity development for rural regions to reduce demand on retrieval services. They could mentioned other mitigation strategies like telehealth that have been shown to reduce the need for aeromedical evacuation.

My only major issue with the article is that the introduction sets up this vital conflict between the need for adaptation measures and the reality that aeromedical transport represents a high carbon footprint demand from the health sector due to air transport. However, while I see this as a crucial area for discussion, discussion of this conflict is not seen elsewhere in the article. Are there any data sets that record the approximate carbon footprint of these flights and could this be highlighted in the data? Or perhaps this could be an area for future exploration? It seems to me that at some point environmental impact data will need to be weighed alongside medical need in the calculus of the need for long range medical transport.

Author Response

Thank you for the generous review.

We concur with reviewer 3 and have added a sentence on the utility of telehealth (line 322).

We also entirely agree with your final point and have finished the second draft Conclusion section addressing this issue front-on.

Round 2

Reviewer 1 Report

Comments and Suggestions for Authors

The revised paper has improved considerably. It has adequately addressed my comments.